# Turn-Level Active Learning for Dialogue State Tracking

**Zihan Zhang[1], Meng Fang[2], Fanghua Ye[3], Ling Chen[1], Mohammad-Reza Namazi-Rad[4]**

[1]University of Technology Sydney  [2]University of Liverpool
[3]University College London  [4]University of Wollongong

`Zihan.Zhang-5@student.uts.edu.au`, `Meng.Fang@liverpool.ac.uk`
`fanghua.ye.19@ucl.ac.uk`, `Ling.Chen@uts.edu.au`, `mrad@uow.edu.au`

## Abstract

Dialogue state tracking (DST) plays an important role in task-oriented dialogue systems. However, collecting a large amount of turn-by-turn annotated dialogue data is costly and inefficient. In this paper, we propose a novel *turn-level* active learning framework for DST to actively select turns in dialogues to annotate. Given the limited labelling budget, experimental results demonstrate the effectiveness of selective annotation of dialogue turns. Additionally, our approach can effectively achieve comparable DST performance to traditional training approaches with significantly less annotated data, which provides a more efficient way to annotate new dialogue data[1].

## 1 Introduction

Dialogue state tracking (DST) constitutes an essential component of task-oriented dialogue systems. The task of DST is to extract and keep track of the user's intentions and goals as the dialogue progresses (Williams et al., 2013). Given the dialogue context, DST needs to predict all *(domain-slot, value)* at each turn. Since the subsequent system action and response rely on the predicted values of specified domain-slots, an accurate prediction of the dialogue state is vital.

Despite the importance of DST, collecting annotated dialogue data for training is expensive and time-consuming, and how to efficiently annotate dialogue is still challenging. It typically requires human workers to manually annotate dialogue states (Budzianowski et al., 2018) or uses a Machines Talking To Machines (M2M) framework to simulate user and system conversations (Shah et al., 2018). Either way, every turn in the conversation needs to be annotated because existing DST approaches are generally trained in a fully supervised manner, where turn-level annotations are re-

quired. However, if it is possible to find the most informative and valuable turn in a dialogue to label, which enables the training of a DST model to achieve comparable performance, we could save the need to annotate the entire dialogue, and could efficiently utilize the large-scale dialogue data collected through API calls.

Active Learning (AL) aims to reduce annotation costs by choosing the most important samples to label (Settles, 2009; Fang et al., 2017; Zhang et al., 2022). It iteratively uses an acquisition strategy to find samples that benefit model performance the most. Thus, with fewer labelled data, it is possible to achieve the same or better performance. AL has been successfully applied to many fields in natural language processing and computer vision (Schumann and Rehbein, 2019; Casanova et al., 2020; Ein-Dor et al., 2020; Hu and Neubig, 2021). However, the adoption of AL in DST has been studied very rarely. Xie et al. (2018) have studied to use AL to reduce the labelling cost in DST, using a *dialogue-level* strategy. They select a batch of dialogues in each AL iteration and label the entire dialogues (e.g., every turn of each dialogue), which is inefficient to scale to tremendous unlabelled data. To our knowledge, *turn-level* AL remains unstudied for the task of DST.

Furthermore, existing DST approaches (Wu et al., 2019; Heck et al., 2020; Tian et al., 2021; Zhu et al., 2022) treat each dialogue turn as a single, independent training instance with no difference. In fact, in the real-world, utterances in a dialogue have different difficulty levels (Dai et al., 2021) and do not share equal importance in a conversation. For example, in Fig.1, turn-1 is simple and only contains a single domain-slot and value (i.e., *hotel-name=Avalon*), while turn-2 is more complex and generates three new domain-slots, i.e., *hotel-book day, hotel-book people, hotel-book stay*. Given the limited labelling budget, it is an obvious choice to label turn-2 instead of turn-1 since the former is

---

[1]Code and data are available at `https://github.com/hyintell/AL-DST`.

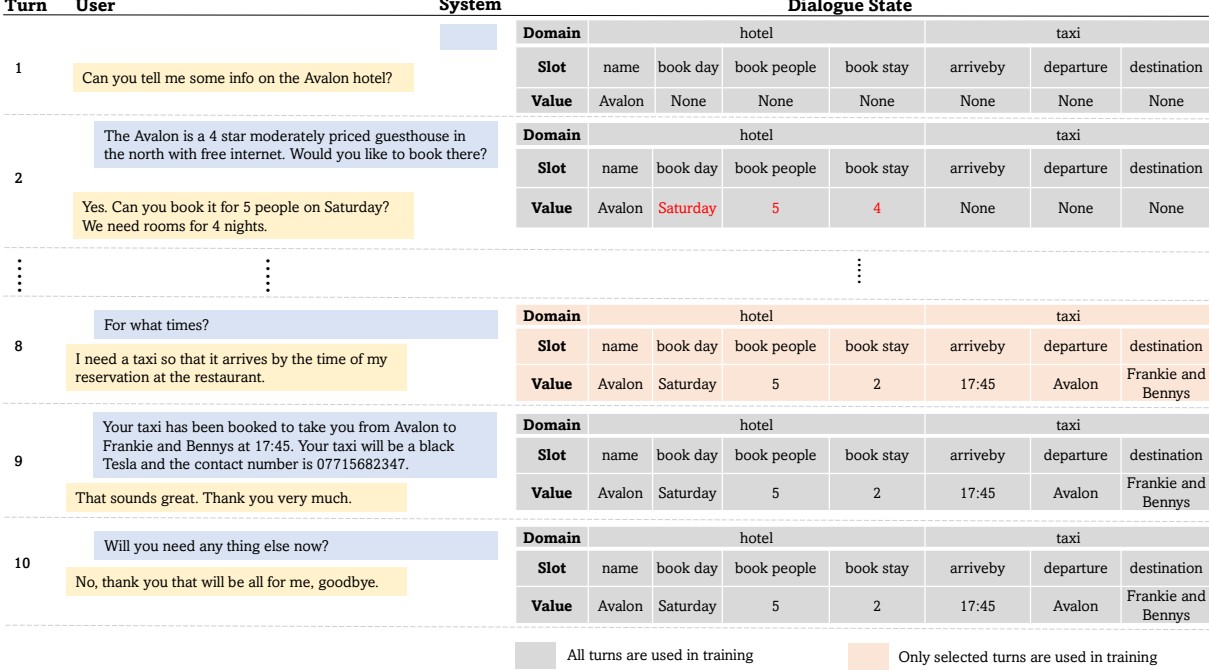

| Turn | User | System | Dialogue State | | | | | | | |
|---|---|---|---|---|---|---|---|---|---|---|

Figure 1: An example of DST from the MultiWOZ dataset (Budzianowski et al., 2018). Utterances at the left and the right sides are from user and system, respectively. Orange color denotes only the selected turn is used in the weakly-supervised training setup. Only two domains (e.g *hotel, taxi*) are shown due to space limitation. (best viewed in color).

more informative[2]. In addition, we observe that the complete states of the dialogue session are updated at turn-8, while turn-9 and turn-10 simply show humans' politeness and respect without introducing any new domain-slots. Therefore, while the "last turn" has been studied before (Lin et al., 2021a), it is often not the case that only the last turn of a dialogue session generates summary states. Using redundant turns such as turn-9 and turn-10 for training not only requires additional labelling but also possibly distracts the DST model since it introduces irrelevant context information, thus hindering the overall performance (Yang et al., 2021).

Built on these motivations, we investigate a practical yet rarely studied problem: *given a large amount of unlabelled dialogue data with a limited labelling budget, how can we annotate the raw data more efficiently and achieve comparable DST performance?* To this end, we propose a novel turn-level AL framework for DST that selects the most valuable turn from each dialogue for labelling and training. Experiments on MultiWOZ 2.0 and 2.1 show that our approach outperforms two strong DST baselines in the weakly-supervised scenarios and achieves comparable DST performance with

significantly less annotated data, demonstrating both effectiveness and data efficiency. We summarize the main contributions of our work as follows:

- We propose a novel model-agnostic *turn-level* Active Learning framework for dialogue state tracking, which provides a more efficient way to annotate new dialogue data. To our best knowledge, this is the first attempt to apply turn-level AL to DST.
- The superiority of our approach is twofold: firstly, our approach strategically selects the most valuable turn from each dialogue to label, which largely saves annotation costs; secondly, using significantly reduced annotation data, our method achieves the same or better DST performance under the weakly-supervised setting.
- We investigate how turn-level AL can boost the DST performance by analyzing the query sizes, base DST models, and turn selection strategies.

## 2 Related Work

### 2.1 Dialogue State Tracking

Dialogue state tracking is an essential yet challenging task in task-oriented dialogue systems (Williams et al., 2013). Recent state-of-the-art DST

---

[2]Here, *informative* refers to the turn that has more valid dialogue states.

models (Wu et al., 2019; Kim et al., 2020; Heck et al., 2020; Ye et al., 2021; Tian et al., 2021; Lee et al., 2021; Zhu et al., 2022; Hu et al., 2022) using different architectures and mechanisms have achieved promising performance on complex multi-domain datasets (Budzianowski et al., 2018; Eric et al., 2020). However, they are generally trained with extensive annotated data, where each dialogue turn requires comprehensive labelling.

To mitigate the cost of dialogue annotation, some works train DST models on existing domains and perform few-shot learning to transfer prior knowledge to new domains (Wu et al., 2019; Zhou and Small, 2019), while others further improve transfer learning by pre-training extensive heterogeneous dialogue corpora using constructed tasks (Wu et al., 2020; Peng et al., 2021; Lin et al., 2021b; Su et al., 2022). Recently, Liang et al. (2021); Lin et al. (2021a) propose a weakly-supervised training setup, in which only the last turn of each dialogue is used. Despite the promising results, we have shown the potential drawbacks of using the last turns in Section 1. In contrast, in this work, we consider the differences between the turns and strategically select the turn that benefits the DST model the most from a dialogue for training.

## 2.2 Active Learning

Active Learning uses an acquisition strategy to select data to minimize the labelling cost while maximizing the model performance (Settles, 2009). While AL has been successfully used in many fields, such as image segmentation (Casanova et al., 2020), named entity recognition (Shen et al., 2017), text classification (Schumann and Rehbein, 2019), and machine translation (Zeng et al., 2019; Hu and Neubig, 2021), rare work has attempted to apply AL to DST. Moreover, recently proposed AL acquisition methods are, unfortunately, not applicable to turn-level DST since they are designed for specific tasks or models. For instance, BADGE (Ash et al., 2019) calculates gradient embeddings for each data point in the unlabelled pool and uses clustering to sample a batch, whereas we treat each turn within a dialogue as a minimum data unit and only select a single turn from each dialogue; therefore, the diversity-based methods are not applicable to our scenario. ALPS (Yuan et al., 2020) uses the masked language model loss of BERT (Devlin et al., 2019) to measure uncertainty in the downstream text classification task, while CAL (Margatina et al., 2021)

selects contrastive samples with the maximum disagreeing predictive likelihood. Both are designed for classification tasks, so these strategies are not directly applicable. Therefore, studying an AL acquisition strategy that is suitable for DST is still an open question.

## 3 Preliminaries

We formalize the notations and terminologies used in the paper as follows.

**Active Learning (AL)**   AL aims to strategically select informative unlabelled data to annotate while maximizing a model's training performance (Settles, 2009). This paper focuses on pool-based active learning, where an unlabelled data pool is available. Suppose that we have a model $\mathcal{M}$, a small seed set of labelled data $\mathcal{L}$ and a large pool of unlabelled data $\mathcal{U}$. A typical iteration of AL contains three steps: (1) train the model $\mathcal{M}$ using $\mathcal{L}$; (2) apply an acquisition function $\mathcal{A}$ to select $k$ instances from $\mathcal{U}$ and ask an oracle to annotate them; and (3) add the newly labelled data into $\mathcal{L}$.

**Dialogue State Tracking (DST)**   Given a dialogue $D = \{(X_1, B_1), \ldots, (X_T, B_T)\}$ that contains $T$ turns, $X_t$ denotes the dialogue turn consisting of the user utterance and system response at turn $t$, while $B_t$ is the corresponding dialogue state. The dialogue state at turn $t$ is defined as $B_t = \{(d_j, s_j, v_j), 1 \leq j \leq J\}$, where $d_j$ and $s_j$ denote domain (e.g. *attraction*) and slot (e.g. *area*) respectively, $v_j$ is the corresponding value (e.g. *south*) of the domain-slot, and $J$ is the total number of predefined domain-slot pairs. Given the dialogue context up to turn $t$, i.e. $H_t = \{X_1, \ldots, X_t\}$, the objective of DST is to predict the value for each domain-slot in dialogue state $B_t$.

**Labelling**   Suppose that we have selected a turn $t$ from the dialogue $D$ ($1 \leq t \leq T$) to label. An oracle (e.g. human annotator) reads the dialogue history from $X_1$ to $X_t$ and labels the current dialogue state $B_t$. We use the gold training set to simulate a human annotator in our experiments.

**Full vs. Weakly-supervised Training**   Generally, the training dataset for DST is built in the way that each turn in a dialogue (concatenated with all previous turns) forms an individual training instance. That is, the input of a single training instance for turn $t$ is defined as $M_t = X_1 \oplus X_2 \oplus \cdots \oplus X_t$, where $\oplus$ denotes the concatenation of sequences,

and the output is the corresponding dialogue state $B_t$. By providing the entire dialogue utterances from the first turn to turn $t$ to the model, the information from the earlier turns is kept in the dialogue history. Let $\mathcal{D}_D$ be the set of training instances created for the dialogue $D$ and $t$ is the selected turn. Given the example in Fig.1, for full supervision, all turns are used for training (i.e., $\mathcal{D}_D = \{(M_1, B_1), \ldots, (M_T, B_T)\}$), whereas in weakly-supervised training, only the selected turn is used (i.e., $\mathcal{D}_D = \{(M_t, B_t)\}$).

## 4 Active Learning for Dialogue State Tracking

In this section, we first define our turn-level AL-based DST framework, followed by the turn selection strategies.

### 4.1 Turn-Level AL for DST

**Framework.** Our turn-level AL-based DST consists of two parts. First, we use AL to model the differences between turns in a dialogue and find the turn that is the most beneficial to label. The pseudocode of this step is shown in Algo. 1. Second, after acquiring all labelled turns, we train a DST model as normal and predict the dialogue states for all turns in the test set for evaluation, as described in Section 3. In this paper, we assume the training set is unlabelled and follow the cold-start setting (Algo. 1 Line 4), where the initial labelled data pool $\mathcal{L} = \emptyset$. We leave the warm-start study for future work.

**Active Learning Loop.** In each iteration, we first randomly sample $k$ dialogues from the unlabelled pool $\mathcal{U}$. Then, we apply a turn acquisition function $\mathcal{A}$ and the intermediate DST model trained from the last iteration to each dialogue $D$ to select an unlabelled turn (Algo. 1 Line 10). It is noteworthy that we consider each turn within a dialogue as a minimum data unit to perform query selection. This is significantly different from Xie et al. (2018), where they select a few dialogues from the unlabelled pool and label all turns as the training instances. Orthogonal to Xie et al. (2018)'s work, it is possible to combine our turn-level strategy with dialogue-level AL. However, we leave it as future work because the AL strategies to select dialogues and turns could be different to achieve the best performance. In this work, we focus on investigating the effectiveness of AL strategies for turn selection.

To avoid overfitting, we re-initialize the base DST model and re-train it on the current accumulated labelled data $\mathcal{L}$. After $R$ iterations, we acquire the final training set $\mathcal{L}$.

---

**Algorithm 1** Turn-level AL for DST

---

**Require:** Initial DST model $\mathcal{M}$, unlabelled dialogue pool $\mathcal{U}$, labelled data pool $\mathcal{L}$, number of queried dialogues per iteration $k$, acquisition function $\mathcal{A}$, total iterations $R$
1: **if** $\mathcal{L} \neq \emptyset$ **then**
2:      $\mathcal{M}_0 \leftarrow$ Train $\mathcal{M}$ on $\mathcal{L}$     ▷ Warm-start
3: **else**
4:      $\mathcal{M}_0 \leftarrow \mathcal{M}$               ▷ Cold-start
5: **end if**
6: **for** iterations $r = 1 : R$ **do**
7:      $\mathcal{X}_r = \emptyset$
8:      $\mathcal{U}_r \leftarrow$ Random sample $k$ dialogues from $\mathcal{U}$
9:      **for** dialogue $D \in \mathcal{U}_r$ **do**
10:          $X \leftarrow \mathcal{A}(\mathcal{M}_{r-1}, D)$    ▷ Select a turn
11:          $\mathcal{X}_r = \mathcal{X}_r \cup \{X\}$
12:      **end for**
13:      $\mathcal{L}_r \leftarrow$ Oracle labels $\mathcal{X}_r$
14:      $\mathcal{L} = \mathcal{L} \cup \mathcal{L}_r$
15:      $\mathcal{U} = \mathcal{U} \setminus \mathcal{U}_r$
16:      $\mathcal{M}_r \leftarrow$ Re-initialize and re-train $\mathcal{M}$ on $\mathcal{L}$
17: **end for**
18: **return** $\mathcal{L}$         ▷ The final training set

---

### 4.2 Turn Selection Strategies

As mentioned in Section 2.2, recently proposed AL acquisition strategies are not applicable to DST. Therefore, we adapt the common uncertainty-based acquisition strategies to select a turn from a dialogue:

**Random Sampling (RS)** We randomly select a turn from a given dialogue. Despite its simplicity, RS acts as a strong baseline in literature (Settles, 2009; Xie et al., 2018; Ein-Dor et al., 2020).

$$X = \text{Random}(M_1, \ldots, M_T) \qquad (1)$$

where $T$ is the total number of turns in the dialogue.

**Maximum Entropy (ME)** (Lewis and Gale, 1994) Entropy measures the prediction uncertainty of the dialogue state in a dialogue turn. In particular, we calculate the entropy of each turn in the dialogue and select the highest one. To do that, we use the base DST model to predict the value of the $j$th domain-slot at turn $t$, which gives us the value

prediction distribution $\mathbf{P}_t^j$. We then calculate the entropy of the predicted value using $\mathbf{P}_t^j$ (Eq.2):

$$\mathbf{e}_t^j = -\sum_{i=1}^{V} \mathbf{p}_t^j[i] \log \mathbf{p}_t^j[i] \qquad (2)$$

$$\mathbf{e}_t = \sum_{j=1}^{J} \mathbf{e}_t^j \qquad (3)$$

$$X = \operatorname{argmax}(\mathbf{e}_1, \ldots, \mathbf{e}_T) \qquad (4)$$

where $V$ is all possible tokens in the vocabulary. We then sum the entropy of all domain-slots as the turn-level entropy (Eq.3) and select the maximum dialogue turn (Eq.4).

**Least Confidence (LC)**    LC typically selects instances where the most likely label has the lowest predicted probability (Culotta and McCallum, 2005). In DST, we use the sum of the prediction scores for all $J$ domain-slots to measure the model's confidence when evaluating a dialogue turn, and select the turn with the minimum confidence:

$$\mathbf{c}_t = \sum_{j=1}^{J} \mathbf{c}_t^j \qquad (5)$$

$$X = \operatorname{argmin}(\mathbf{c}_1, \ldots, \mathbf{c}_T) \qquad (6)$$

where $\mathbf{c}_t^j = \max(\text{logits}_t^j)$ denotes the maximum prediction score of the $j$th domain-slot at turn $t$ and $\text{logits}_t^j$ is the predictive distribution.

## 5 Experiments

### 5.1 Setup

**Datasets.**    We evaluate the weakly-supervised DST performance on the MultiWOZ 2.0 (Budzianowski et al., 2018) and MultiWOZ 2.1 (Eric et al., 2020) datasets[3] as they are widely adopted in DST. We use the same preprocessing as Lin et al. (2021a) and Su et al. (2022), and focus on five domains (i.e. *restaurant, train, hotel, taxi, attraction*). The statistics of the datasets are summarized in Appendix A.

---

[3]We also tried to use the SGD dataset (Rastogi et al., 2020). However, the PPTOD model is already pre-trained on this dataset, making it unsuitable for downstream evaluation. KAGE-GPT2 requires the predefined ontology to build a graph neural network, but SGD does not provide all possible values for non-categorical slots (See Section 8).

**Base DST Model.**    We use **KAGE-GPT2** (Lin et al., 2021a) as the base DST model to implement all experiments. KAGE-GPT2 is a generative model that incorporates a Graph Attention Network to explicitly learn the relationships between domain-slots before predicting slot values. It shows strong performance in both full and weakly-supervised scenarios on MultiWOZ 2.0 (Budzianowski et al., 2018). To show that the effectiveness of our AL framework is not tied to specific base models, we also experiment with an end-to-end task-oriented dialogue model **PPTOD** (Su et al., 2022). PPTOD pre-trained on large dialogue corpora gains competitive results on DST in the low-resource settings. The model training and implementation details are in Appendix B.

### 5.2 Evaluation Metrics

We use **Joint Goal Accuracy (JGA)** to evaluate DST performance, which is the ratio of correct dialogue turns. It is a strict metric since a turn is considered as correct if and only if all the slot values are correctly predicted. Following the community convention, although it is not a distinguishable metric (Kim et al., 2022), we also report **Slot Accuracy (SA)**, which compares the predicted value with the ground truth for each domain-slot at each dialogue turn. Additionally, we define a new evaluation metric, **Reading Cost (RC)**, which measures the number of turns a human annotator needs to read to label a dialogue turn. As shown in Fig.1, to label the dialogue state $B_t$ at turn $t$, a human annotator needs to read through the dialogue conversations from $X_1$ to $X_t$ to understand all the domain-slot values that are mentioned in the dialogue history:

$$\text{RC} = \frac{\sum_{i=1}^{|\mathcal{L}|} \frac{t}{T_{D_i}}}{|\mathcal{L}|} \qquad (7)$$

where $|\mathcal{L}|$ denotes the total number of annotated dialogues and $T_{D_i}$ is the number of turns of the dialogue $D_i$. If all last turns are selected, then $\text{RC} = 1$, in which case the annotator reads all turns in all dialogues to label, resulting high cost. Note that we take JGA and RC as primary evaluation metrics.

### 5.3 Baselines

Our main goal is to use AL to actively select the most valuable turn from each dialogue for training, therefore reducing the cost of labelling the entire dialogues. We evaluate the effectiveness of our

| Training Data | Model | MultiWOZ 2.0 | | | MultiWOZ 2.1 | | |
|---|---|---|---|---|---|---|---|
| | | JGA ↑ | SA ↑ | RC ↓ | JGA ↑ | SA ↑ | RC ↓ |
| | | *Without Active Learning* | | | | | |
| **Full Data (100%)** | PPTOD$_{base}$ | 53.37±0.46 | 97.26±0.02 | 100 | 57.10±0.51 | 97.94±0.02 | 100 |
| | KAGE-GPT2 | 54.86±0.12 | 97.47±0.02 | 100 | 52.13±0.89 | 97.18±0.02 | 100 |
| **Last Turn (14.4%)** | PPTOD$_{base}$-LastTurn | 43.83±1.55 | 96.87±0.06 | 100 | 45.94±0.72 | 97.11±0.04 | 100 |
| | KAGE-GPT2-LastTurn | 50.43±0.23 | 97.14±0.01 | 100 | 49.12±0.13 | 97.05±0.02 | 100 |
| | | *With Active Learning ($k = 2000$)* | | | | | |
| **CUDS (∼14%)**[*] | PPTOD$_{base}$+CUDS | 43.06±0.04 | 96.01±0.02 | 100 | 43.57±1.16 | 96.16±0.01 | 100 |
| | KAGE-GPT2+CUDS | 47.06±1.43 | 96.14±0.07 | 100 | 47.56±1.07 | 96.33 | 100 |
| **Selected Turn (14.4%) (Ours)** | PPTOD$_{base}$+RS | 43.71±0.81 | 96.64±0.08 | 58.73±28.7 | 46.96±0.18 | 96.56±0.06 | **58.55**±28.5 |
| | PPTOD$_{base}$+LC | 45.79±0.35 | 97.06±0.04 | 85.21±19.7 | 47.37±0.32 | 96.97±0.05 | 81.95±24.6 |
| | PPTOD$_{base}$+ME | **46.92**±0.79 | **97.12**±0.05 | **57.37**±32.9 | **48.21**±1.00 | **97.33**±0.12 | 67.68±30.1 |
| | KAGE-GPT2+RS | 50.37±0.52 | 97.11±0.06 | **58.58**±28.7 | 46.98±0.64 | 96.81±0.07 | 58.48±28.5 |
| | KAGE-GPT2+LC | 50.56±0.07 | 97.10±0.01 | 70.51±30.3 | 48.13±0.20 | 96.94±0.01 | 79.41±24.0 |
| | KAGE-GPT2+ME | **51.34**±0.05 | **97.16**±0.05 | 62.58±28.5 | **50.02**±1.10 | **97.13**±0.10 | 71.02±26.7 |

Table 1: The mean and standard deviation of joint goal accuracy (%), slot accuracy (%) and reading cost (%) after the final AL iteration on the test sets. [*]: we re-implement using Xie et al. (2018)'s method. **RS**, **LC** and **ME** are active turn selection methods mentioned in Section 4.2. Note that we take JGA and RC as primary evaluation metrics since SA is indistinguishable (Kim et al., 2022).

approach from two angles. First, we compare DST performance of two settings *without* involving AL to show the benefits that AL brings:

- **Full Data (100%)**: all the turns are used for training, which shows the upper limit of the base DST model performance.
- **Last Turn (14.4%[4])**: following Liang et al. (2021) and Lin et al. (2021a), for each dialogue, only the last turn is used for training.

Second, when using AL, we compare our turn-level framework with the dialogue-level approach:

- **CUDS (∼14%)** (Xie et al., 2018): a dialogue-level method that selects a batch of dialogues in each AL iteration based on the combination of labelling cost, uncertainty, and diversity, and uses all the turns for training. We carefully maintain the number of selected dialogues in each iteration so that the total number of training instances is roughly the same (i.e., $k \simeq 2000$) for a fair comparison.
- **Selected Turn (14.4%)**: we apply Algo.1 and set $\mathcal{U} = 7888$, $\mathcal{L} = \emptyset$, $k = 2000$ and use the turn selection methods mentioned in Section 4.2 to conduct experiments. As a trade-off between computation time and DST performance, here we use $k = 2000$; however, we find that a smaller $k$ tends to have a better performance (Section 6.2). Given $k = 2000$, we have selected 7,888 turns after four rounds, and use them to train a final model.

[4] $14.4\% = \frac{\text{\# turns used}}{\text{\# total turns}} = \frac{7888}{54945}$

## 6 Results & Analysis

### 6.1 Main Results

Due to space limitation, we report the final results after the four AL iterations in Table 1. We present the intermediate results in Fig.2.

**Our *turn-level* AL strategy improves DST performance.** From Table 1, we first observe that, using the same amount of training data (14.4%), our proposed AL approach (i.e. PPTOD$_{base}$+ME and KAGE-GPT2+ME) outperforms the non-AL settings, **Last Turn**, in terms of both joint goal accuracy and slot accuracy. Specifically, compared with PPTOD$_{base}$+LastTurn, our PPTOD$_{base}$+ME significantly boosts the JGA by 3.1% on MultiWOZ 2.0 and 2.3% on MultiWOZ 2.1. KAGE-GPT2+ME also improves its baselines by around 0.9% on both datasets. Compared with the dialogue-level AL strategy **CUDS**, our turn-level methods improve the JGA by a large margin (2.3%∼4.3% on both datasets). Considering that DST is a difficult task (Budzianowski et al., 2018; Wu et al., 2019; Lee et al., 2021), such JGA improvements demonstrate the effectiveness of our turn-level AL framework, which can effectively find the turns that the base DST model can learn the most from.

**Our *turn-level* AL strategy reduces annotation cost.** The reading costs (RC) of PPTOD$_{base}$+ME and KAGE-GPT2+ME drop by a large margin (around 29%∼43%) compared to the Last Turn and CUDS settings, indicating the benefits and necessity of

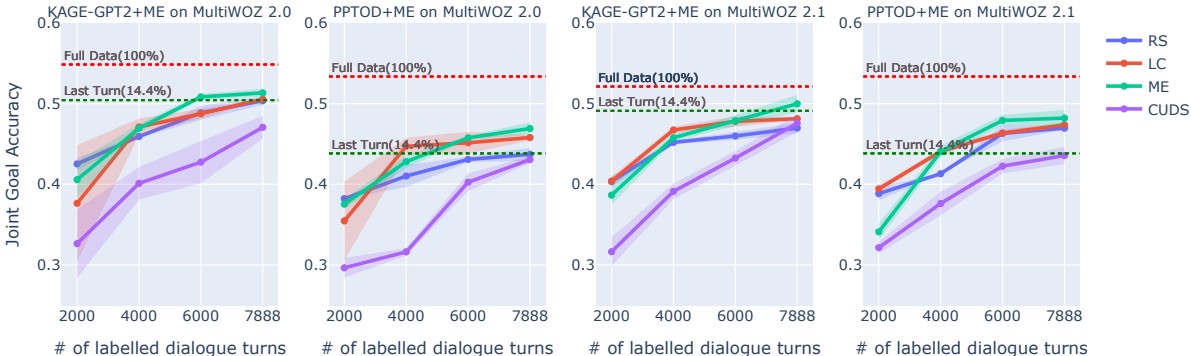

Figure 2: Joint goal accuracy on test sets of AL over four iterations with $k = 2000$ dialogues queried per iteration.

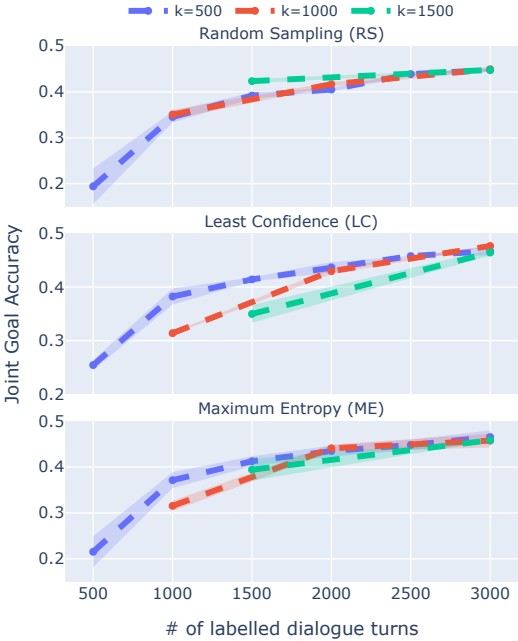

Figure 3: Joint goal accuracy on test sets of KAGE-GPT2 on MultiWOZ 2.0 with $k = 500, 1000, 1500$.

selecting dialogue turns. This significantly saves the annotation cost because a human annotator does not need to read the entire dialogue to label the last turn but only needs to read until the selected turn.

**Our approach uses less annotated data can achieve the same or better DST performance.** To further explore the capability of our AL approach, we plot the intermediate DST performance during the four iterations, as shown in Fig.2. Notably, PPTOD_base with Least Confidence (LC) and Maximum Entropy (ME) turn selection methods surpass the Last Turn baselines at just the second or third iteration on MultiWOZ 2.0 and MultiWOZ 2.1 respectively, showing the large data efficiency of our approach (only 7.3% / 10.9% data are used). This can be explained that PPTOD_base is fine-tuned

on so-far selected turns after each iteration and gains a more robust perception of unseen data, thus tending to choose the turns that are more beneficial to the model. In contrast, KAGE-GPT2 underperforms the Last Turn setting in early iterations, achieving slightly higher accuracy in the final round. Despite this, the overall performance of KAGE-GPT2 is still better than PPTOD_base under the weakly-supervised settings. This is possibly because the additional graph component in KAGE-GPT2 enhances the predictions at intermediate turns and the correlated domain-slots (Lin et al., 2021a). However, when using CUDS, both DST models underperform a lot on both datasets, especially during early iterations. This indicates that the dialogue-level strategy, which does not distinguish the importance of turns in a dialogue, might not be optimal for selecting training data. In Section 6.2, we show that a smaller query size $k$ can achieve higher data efficiency.

## 6.2 Ablation Studies

In this section, we further investigate the factors that impact our turn-level AL framework.

**Effect of Dialogue Query Size.** Theoretically, the smaller size of queried data per AL iteration, the more intermediate models are trained, resulting the better model performance. Moreover, smaller query size is more realistic since the annotation budget is generally limited and there lack enough annotators to label large amount of dialogues after each iteration. To this end, we initialize the unlabelled pool $\mathcal{U}$ by randomly sampling 3,000 dialogues from the MultiWOZ 2.0 training set, and apply our AL framework to KAGE-GPT2, using different query sizes, i.e., $k = 500, 1000, 1500$, which leads to $6, 3, 2$ rounds respectively.

From Fig.3, we first observe that smaller $k$ im-

proves the intermediate DST performance: when $k = 500$, both LC and ME strategies boost the accuracy by a large margin at the second iteration than $k = 1000$, and at the third iteration than $k = 1500$. This suggests that, with the same number of training data, the multiple-trained DST model gains the ability to have a more accurate perception of the unseen data. By calculating the prediction uncertainty of the new data, the model tends to choose the turns that it can learn the most from. In contrast, RS chooses a random turn regardless of how many AL rounds, therefore does not show the same pattern as LC and ME. Finally, we find a smaller $k$ tends to achieve higher data efficiency when using LC and ME strategies. It is clear from the figure that $k = 500$ uses the least data when reaching the same level of accuracy. However, the drawback of a smaller query size is that it increases overall computation time as more intermediate models have to be trained. We provide a computational cost analysis in Section 6.3.

**Effect of Base DST Model.** It is no doubt that the base DST model is critical to our turn-level AL framework as it directly determines the upper and lower limit of the overall performance. However, we are interested to see how our approach can further boost the performance of different DST models. We randomly sample $\mathcal{U} = 500$ dialogues from the MultiWOZ 2.0 training set and set the query size $k = 100$ for both models. As shown in Fig.4, we also report the results of the two models using the non-AL strategy of Last Turn, which can be considered as the lower performance baselines.

We first confirm that both PPTOD$_{base}$ and KAGE-GPT2 outperform their Last Turn baselines after applying our AL framework, demonstrating both data efficiency and effectiveness of our approach. Secondly, we notice that PPTOD$_{base}$ achieves comparable accuracy in the first two rounds, while KAGE-GPT2 nearly stays at 0 regardless of the turn selection methods, showing the superiority of PPTOD$_{base}$ under the extreme low-resource scenario. This is possibly because PPTOD$_{base}$ is pre-trained on large dialogue corpora thus gains few-shot learning ability (Su et al., 2022), whereas only 200 training data are not enough for KAGE-GPT2 to be fine-tuned. However, in the later iterations, the performance of KAGE-GPT2 grows significantly, especially when using the ME strategy, eventually reaching the same level as PPTOD$_{base}$. In contrast, the accu-

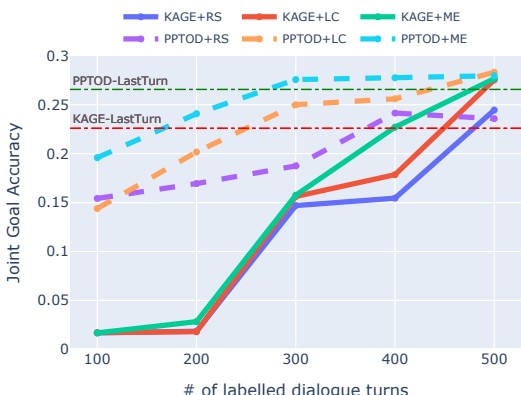

Figure 4: Joint goal accuracy on test sets of KAGE-GPT2 and PPTOD$_{base}$ on MultiWOZ 2.0 with $k = 100$. Results are averaged over three runs.

| Method | KAGE-GPT2 | PPTOD$_{base}$ |
|--------|-----------|----------------|
| LC | $76.51_{\pm 24.7}$ | $81.13_{\pm 22.3}$ |
| ME | $68.18_{\pm 29.1}$ | $58.68_{\pm 31.5}$ |

Table 2: Reading Cost (RC) (%) of different turn selection methods. The lower the better.

racy of PPTOD$_{base}$ increases slowly, indicating the model gradually becomes insensitive to the newly labelled data.

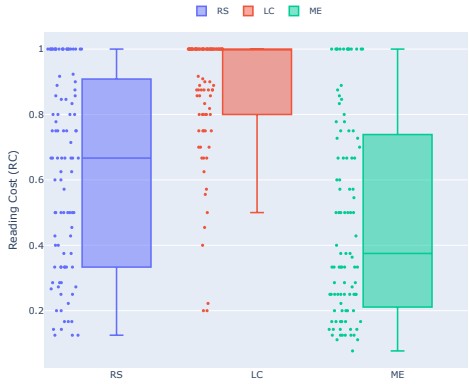

Figure 5: Visualization of the turns selected by PPTOD$_{base}$ at the final round ($k = 100$). ME reduces RC the most.

**Effect of Turn Selection Strategy.** From Fig.2, while both ME and LC improve over the RS baseline, ME does not consistently outperform LC during AL iterations in terms of the joint goal accuracy, and vice versa. However, as shown in Table 1, LC results in a higher Reading Cost (RC) than ME, which means LC tends to select latter half of turns in dialogues. Conversely, ME significantly reduces RC in the last iteration (Fig.5; more in Appendix C) and is consistently better than LC and RS for

Table 3: Example (MUL0295) of the selected turn (marks by ✓) by PPTOD$_{base}$ using ME and LC.

| Dialogue MUL0295 | ME | LC |
|---|---|---|
| [S]: 
 Turn 1   [U]: i am looking for an expensive place to dine in the centre of town. 
 *State: {restaurant-area=centre, restaurant-pricerange=expensive}* | | |
| [S]: great kymmoy is in the centre of town and expensive. 
 Turn 2   [U]: i want to book a table for 3 people at 14:00 on Saturday. 
 *State: {restaurant-book day=saturday, restaurant-book people=3, restaurant-book time=14:00}* | | |
| [S]: booking was successful. the table will be reserved for 15 minutes. reference number is: vbpwad3j. 
 Turn 3   [U]: thank you so much. i would also like to find a train to take me to kings lynn by 10:15. 
 *State: {train-destination=kings lynn, train-arriveby=10:15}* | | ✓ |
| [S]: there are 35 departures with those criteria. what time do you want to leave? 
 Turn 4   [U]: the train should arrive by 10:15 please on sunday please. 
 *State: {train-day=sunday}* | ✓ | |
| [S]: how many tickets will you need? 
 Turn 5   [U]: just 1 ticket. i will need the train id, cost of ticket and exact departure time as well. 
 *State: {}* | | |
| [S]: there is a train arriving in kings lynn on sunday at 09:58. it departs at 09:11 and costs 7.84 pounds. the train id is tr6088. 
 Turn 6   [U]: great! that s all i needed. thanks a lot for the help. 
 *State: {}* | | |

| Method | # of Training data (%) ↓ | JGA ↑ | RC ↓ | Runtime (hour) ↓ |
|---|---|---|---|---|
| Full data | 21072 (100%) | 46.7 | 100 | 2.3 |
| Last Turn | 3000 (14.2%) | 41.4 | 100 | 0.6 |
| ME | 3000 (14.2%) | 44.3 | 59.3 | 1.6 |

Table 4: Computational cost comparison using KAGE-GPT2 on MultiWOZ 2.0 with $\mathcal{U} = 3000$ and $k = 1000$.

| Method | Total Annotation Cost ($) ↓ |
|---|---|
| Full Dialogue | $z * (T * x + T * y)$ |
| Last Turn | $z * (T * x + 1 * y)$ |
| Selected Turn (Ours) | $z * (t * x + 1 * y)$, where $1 \leq t \leq T$ |

Table 5: Annotation cost estimation comparison of different methods.

both DST models (Fig.4), which demonstrates the effectiveness of ME under small query size $k$. We report their RC in Table 2, which also confirms that ME saves reading costs than LC. An example of the turns selected by ME and LC in a dialogue is shown in Table 3, more examples in Appendix D.

## 6.3 Cost Analysis

Our AL-based method saves annotation costs and achieves comparable DST performance with traditional methods at the expense of increased computation time. In this section, we conduct a cost analysis, including computation and annotation costs. We initialize the unlabelled pool $\mathcal{U}$ by randomly sampling 3,000 dialogues from the MultiWOZ 2.0 training set, and apply our AL framework to KAGE-GPT2, and set the query size as $k = 1000$. As shown in Table 4, our method improves JGA and RC than the Last Turn baseline, but with an increased runtime since our method requires three rounds of iteration.

Due to a lack of budget, we are unable to employ human annotators to evaluate the actual annotation cost. Instead, we conduct a theoretical cost analysis

to show the potential cost reduction of our method. Suppose a dialogue $D$ has $T$ turns in total, and it takes $x$ minutes for a human annotator to read each turn (*i.e.*, reading time), $y$ minutes to annotate a single turn (*i.e.*, annotating time), $z$ dollars per minute to hire a human annotator. Assuming our proposed method selects the $t$th ($1 \leq t \leq T$) turn to annotate. The total annotation cost, including the reading time and annotating time of three methods, are listed in Table 5. Since the Full Dialogue baseline takes each accumulated turn as a training instance (Section 3), it requires the highest annotation cost. Our method only annotates a single turn per dialogue, the same as the Last Turn baseline. Therefore, the annotation cost lies in the selected turn $t$, which is measured by RC in our experiments. As shown in Table 1 and discussed in Section 6.1, our method generally saves RC by a large margin (around 29%∼43% across different models) compared to the Last Turn baseline and saves more compared to the Full data setting. Therefore, from a theoretical cost estimation point of view, our proposed method can save annotation costs while maintaining DST performance.

## 7 Conclusion

This paper tackles the practical dialogue annotation problem by proposing a novel turn-level AL framework for DST, which strategically selects the most valuable turn from each dialogue for labelling and training. Experiments show that our approach outperforms strong DST baselines in the weakly-supervised scenarios and achieves the same or better joint goal and slot accuracy with significantly less annotated data. Further analysis are conducted to investigate the impact of AL query sizes, base DST models and turn selection methods.

## 8 Limitations

We acknowledge the limitations of this paper as follows.

First, our AL approach adds extra computation time compared to directly training a DST model using only the last turns of dialogues. A smaller query size (e.g., $k$) may further increase the runtime as more intermediate models have to be trained. That is, we achieved similar or even better DST performance with significantly reduced annotation data at the cost of increased computation time. Therefore, the trade-off between computational cost, DST performance, and annotation cost needs to be well-determined.

Second, we are unable to employ human annotators to evaluate the actual cost due to a lack of budget. In practice, the number of annotators required depends on the financial budget, project timeline, and the proficiency of annotators. Estimating the exact number of annotators and the annotation cost is challenging. As a mitigation, we provide a theoretical cost analysis in Section 6.3. However, it is a rough estimation and may not reflect the actual cost.

Third, our experiments are limited to the Multi-WOZ 2.0 (Budzianowski et al., 2018) and Multi-WOZ 2.1 (Eric et al., 2020) datasets. We also tried to use the SGD dataset (Rastogi et al., 2020). However, the PPTOD model is already pre-trained on this dataset, making it unsuitable for downstream evaluation. KAGE-GPT2 requires the predefined ontology (i.e., the all possible domain-slot value pairs in the dataset) to build a graph neural network, but SGD does not provide all possible values for non-categorical slots. For example, MultiWOZ has all possible values predefined for the non-categorical domain-slot *train-arriveBy*, while SGD does not have it since it is innumerable. Our AL framework is built upon the base DST model and thus suffers the same drawbacks; we may try other DST models and datasets in the future.

## Acknowledgements

This work is supported by TPG Telecom. We would like to thank anonymous reviewers for their valuable comments.

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

## A  Datasets Statistics

|  |  | MultiWOZ2.0 | MultiWOZ2.1 |
|---|---|---|---|
| Train | # Dialogues | 7888 | 7888 |
|  | # Domains | 5 | 5 |
|  | # Slots | 30 | 30 |
|  | # Total turns | 54945 | 54961 |
|  | # Last turns | 7888 | 7888 |
|  | # Avg. turns per dialogue | 6.97 | 6.97 |
|  | # Max turns per dialogue | 22 | 22 |
|  | # Min turns per dialogue | 1 | 1 |
| Validation | # Dialogues | 1000 | 1000 |
|  | # Total turns | 7374 | 7374 |
| Test | # Dialogues | 1000 | 999 |
|  | # Total turns | 7372 | 7368 |

Table 6:  Statistics of the datasets in the experiments.

## B  Configuration Details

We use the official release of KAGE-GPT2[5] (Lin et al., 2021a) and PPTOD[6] (Su et al., 2022) to implement our turn-level AL framework.

**KAGE-GPT2**  We use the `L4P4K2-DSGraph` model setup and follow its sparse supervision (last turn) hyperparameter settings. Specifically, the loaded pre-trained GPT-2 model has 12 layers, 768 hidden size, 12 heads and 117M parameters, which is provided by HuggingFace[7]. AdamW optimizer with a linear decay rate $1 \times 10^{-12}$ is used when training. The GPT-2 component and the graph component are jointly trained, with the initial learning rates are $6.25 \times 10^{-5}$ and $8 \times 10^{-5}$ respectively. The training batch size used is 2, while the batch size for validation and evaluation is 16.

**PPTOD**  We use the released `base` checkpoint, which is initialized with a T5-base model with around 220M parameters. $PPTOD_{base}$ is pre-trained on large dialogue corpora, for more details, we refer readers to the original paper. When training, Adafactor optimizer is used and the learning rate is $1 \times 10^{-3}$. Both training, validation, and evaluation batch size used is 4.

**Turn Selection**  During each AL iteration, we use the trained model from the last iteration to evaluate all the turns within a dialogue and then select a turn based on the acquisition strategy.

**Training**  At the end of each iteration, we re-initialize a new pre-trained GPT-2 model for KAGE-GPT2 or re-initialize a new model from the released pre-trained base checkpoint for PPTOD, and then train the model as usual with all current accumulated labelled turns. We train the DST model for 150 epochs using the current accumulated labelled pool $\mathcal{L}$, and early stop when the performance is not improved for 5 epochs on the validation set. Importantly, instead of using the full 7,374 validation set, we only use the last turn of each dialogue to simulate the real-world scenario, where a large amount of annotated validation set is also difficult to obtain (Perez et al., 2021). However, we use the full test set when evaluating.

## C  Visualization of Selected Turns

To clearly compare the reading costs of different turn selection methods, we visualize the distributions of the selected turns at the final round for the setting in Section 6.2, as shown in Fig.5 and Fig.6. A dot means a selected turn from a dialogue, while the ends of the box represent the lower and upper quartiles, and the

---

[5]https://github.com/LinWeizheDragon/Knowledge-Aware-Graph-Enhanced-GPT-2-for-Dialogue-State-Tracking

[6]https://github.com/awslabs/pptod

[7]https://huggingface.co/models

median (second quartile) is marked by a line inside the box. A higher RC means the turn is selected from the second half of the conversation (RC = 1 means the last turn is selected); thus, a human annotator needs to read most of the conversation to label its state, which is more costly. From the figures, overall, RS distributes randomly, while ME has a much lower reading cost than LC, especially for PPTOD$_{base}$.

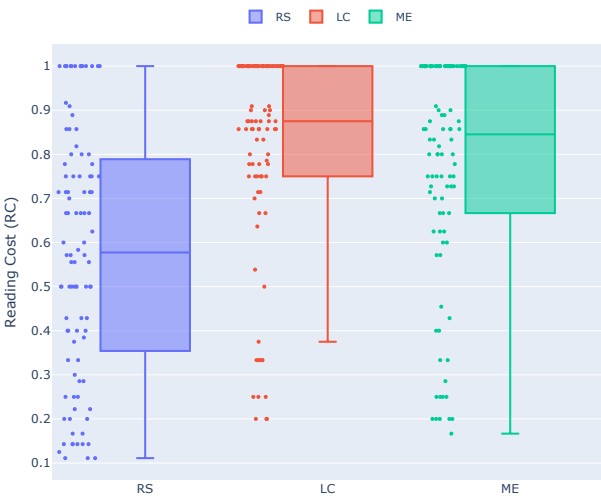

Figure 6: Visualization of the turns selected by KAGE-GPT2 at the final round ($k = 100$).

## D    Example of Selected Turns

Table 3, Table 7 and Table 8 present the examples of selected turns by ME and LC using PPTOD$_{base}$ from MultiWOZ 2.0. [S] and [U] denote the system and user utterance respectively, while *State* represents the dialogue states that are mentioned at the current turn. ✓marks the selected turn by the strategy and is the only turn in the dialogue used for training. Although not always the case, we can see that both ME and LC can select the earliest turn that summarizes the entire dialogue, which not only saves the need to read through the whole conversation but also keeps the valuable context information intact as much as possible. However, still, a more suitable AL query strategy for DST is worthy of being studied.

Table 7: Example (MUL1068) of the selected turn by PPTOD$_{base}$ using ME and LC.

| Dialogue MUL1068 | | ME | LC |
|---|---|---|---|
| Turn 1 | [S]:
[U]: i am looking for a gueshouse to stay in during my visit. i would like for it to be in the east section of town.
*State: {hotel-area=east, hotel-type=guesthouse}* | | |
| Turn 2 | [S]: there are 6 entries, do you have any other preferences in hotel choice?
[U]: i would like a 4 star place that's on the pricier side.
*State: {hotel-pricerange=expensive, hotel-stars=4}* | | |
| Turn 3 | [S]: there are no 4 star expensive guesthouses on the east side of town. would you be interested in a cheap or moderate -ly priced 1 instead?
[U]: i would like 1 that is cheap then.
*State: {hotel-pricerangen=cheap}* | | |
| Turn 4 | [S]: autumn house is cheap and has free internet and parking . would you like me to book a reservation for you?
[U]: yes please. can you book the reservation for saturday for 5 guests for a 2 night stay. thanks.
*State: {hotel-book day=saturday, hotel-book people=5, hotel-book stay=2}* | | |
| Turn 5 | [S]: sorry, there are no rooms available for your specification -s. do you want to try a different day or a shorter stay?
[U]: try for 1 night instead. if that works, can i have the reference number?
*State: {hotel-book stay=1}* | | |
| Turn 6 | [S]: how about the autumn house for 1 night starting saturday and your reference number is svciwlwa.
[U]: i also need to find something called gonville and caius college, can you help?
*State: {attraction-name=gonville and caius college}* | ✓ | |
| Turn 7 | [S]: gonville and caius college is located in the center of town on trinity street . anything else i can do for you?
[U]: can i get the postcode ?
*State: {}* | | |
| Turn 8 | [S]: no problem , the postal code is cb21ta. did you need the phone number as well?
[U]: no thanks i am all set . thank you for your help today.
*State: {}* | | ✓ |

Table 8: Example (PMUL2281) of the selected turn by PPTOD$_{base}$ using ME and LC.

| Dialogue PMUL2281 | ME | LC |
|---|:---:|:---:|
| [S]:
Turn 1  [U]: can you help me find a place to dine?
*State: {}* | | |
| [S]: sure! what type of food would you like to eat in what area?
Turn 2  [U]: i would like some north indian food that is expensive, in the south.
*State: {restaurant-food=north indian}* | | |
| [S]: we do not have any north indian restaurant -s, though we do have many indian restaurant -s.
Turn 3  [U]: indian food would be fine then, can you tell me the name?
*State: {restaurant-pricerange=expensive, restaurant-food=indian}* | | |
| [S]: there are several indian restaurant -s, may i suggest the golden curry in the centre. it sounds like just what you are looking for.
Turn 4  [U]: i was actually hoping for a restaurant in the south. are there any available?
*State: {restaurant-area=south}* | | ✓ |
| [S]: i found 1 called taj tandoori. want more information?
Turn 5  [U]: more information please
*State: {}* | | |
| [S]: it is expensive and in the south.
Turn 6  [U]: can i get the postcode for that please?
*State: {}* | ✓ | |
| [S]: sure! the post code is cb17aa .
Turn 7  [U]: thanks! i am also looking for a nightclub. i'll need the postcode, please?
*State: {attraction-type=nightclub}* | | |
| [S]: the ballare is a nightclub in the centre of town. the entrance fee is 5 pounds.
Turn 8  [U]: i'll try that. what s the postcode please?
*State: {}* | | |
| [S]: their postcode is cb23na. can i help you with anything else today?
Turn 9  [U]: no thanks. that was all i needed today. goodbye.
*State: {}* | | |