# OpenReview forum: "Turn-Level Active Learning for Dialogue State Tracking"
_EMNLP/2023/Conference — EMNLP 2023 Main_

### Official Review · Reviewer_ttCV · 2023-07-31

**Soundness:** 4

**Excitement:**

4: Strong: This paper deepens the understanding of some phenomenon or lowers the barriers to an existing research direction.

**Paper Topic And Main Contributions:**

The paper attempts to tackle a very realistic problem in DST - annotating dialogue turns in turn level is highly expensive. Even with the last-turn annotation strategy proposed in literature (the annotator annotates the last turn instead of the full dialogue), the annotator must read through all turns in a dialogue in order to annotate all dialogue states correctly. This paper presents a turn-level selection strategy based on active learning to significantly reduce annotation costs while maintaining the dialogue state tracking performance. This topic is rarely investigated in current literature, and it shows great potential in reducing the required resources for annotating dialogue states for building advanced DST systems.

**Questions For The Authors:**

One underlying motivation for using the last-turn annotation scheme is that such data could be easily accessed from commercial service data [1]. It can be easy to obtain the actions done by the customer service staff after a phone call. It’d be appreciated if this case can be discussed and compared to the proposed approach.

[1] Weizhe Lin, Bo-Hsiang Tseng, and Bill Byrne. 2021. Knowledge-aware graph-enhanced GPT-2 for dialogue state tracking.

**Reasons To Accept:**

1. Combining active learning to select samples for annotation with the aid of the model being trained is a novel idea, which is rarely explored in literature.
2. The experiments show that with the proposed approach, several recent advanced DST systems can achieve performance comparable to training with full data and weakly supervised data. In particular, Maximum Entropy (ME) achieves better performance than training with last-turn annotations while reducing the RC (Reading Cost) by ~40%, which means annotators do not have to read the full dialogue. This can greatly reduce the workload of annotators.
3. The writing is fluent and clear. The presentation is good.
4. The experiments are complete. Two different models and two datasets were experimented, demonstrating that to some extent this approach is generalisable. I found the analyses interesting, especially those about Reading Cost.

**Reasons To Reject:**

1. Some more discussions regarding the recent progress in DST will be appreciated. MultiWOZ is a traditional DST dataset, while recently the Schema-guided DST dataset is actively investigated. This new setting challenges the few-shot/zero-shot generalisability of DST systems on new services/domains. Though not required in this work, it would be interesting to see a discussion on how this approach could be generalised to/used in the SGDST setting. -- after discussion, the authors promised to add some experiments related to this question.
2. It remains unclear how this approach can be used in practice. Imagine that we want to train a new DST system and we want to annotate domain-specific data for this purpose, what steps are required (in practice) to combine annotators with the on-the-fly training process? Does the model training run in the background while a batch of annotators do async-ed annotation? -- after discussion, the authors promised to add some discussion related to this question.
3. The two datasets being evaluated are very similar (v2.0 and v2.1). It would be better if a third dataset can be evaluated.

**Reproducibility:**

4: Could mostly reproduce the results, but there may be some variation because of sample variance or minor variations in their interpretation of the protocol or method.

**Reviewer Confidence:**

5: Positive that my evaluation is correct. I read the paper very carefully and I am very familiar with related work.

---

> ### Author Rebuttal · Authors · 2023-08-29
>
> Dear Reviewer ttCV,
>
> We appreciate your valuable time, review and suggestions. We want to address your concerns in detail as follows.
>
> > Q1. Some more discussions regarding the recent progress in DST will be appreciated. MultiWOZ is a traditional DST dataset, while recently the Schema-guided DST dataset is actively investigated. This new setting challenges the few-shot/zero-shot generalisability of DST systems on new services/domains. Though not required in this work, it would be interesting to see a discussion on how this approach could be generalised to/used in the SGDST setting.
>
> **A1:** Thanks for your suggestion.
> Our proposed method can be directly applied to the SGD dataset.
> The SGD dataset captures real-world scenarios and contains unseen domains and services in the evaluation set for zero-shot performance evaluation. Using the SGD dataset, it might be interesting to see how our proposed framework can benefit unseen domains. We will add experiments on the SGD dataset when we are allowed to modify our paper.
>
> > Q2. It remains unclear how this approach can be used in practice. Imagine that we want to train a new DST system and we want to annotate domain-specific data for this purpose, what steps are required (in practice) to combine annotators with the on-the-fly training process? Does the model training run in the background while a batch of annotators do async-ed annotation?
>
> **A2:** As shown in Algo1 (L234), following the conventional active learning practices, this process runs iteratively. Firstly, we can start from a base DST model (such as PPTOD, warm start), or a pre-trained LM (such as GPT-2, cold start). Then, given a batch of unannotated dialogues, we use our method to select a turn for each dialogue. Thirdly, we ask annotators to read the dialogue until the selected turn to collect this batch of annotated dialogue data. Finally, we use the collected annotated data to fine-tune the DST model, conduct evaluation and record performance. In the next loop, we start with the latest DST model and annotate the next batch of raw dialogues.
>
> > Q3. The two datasets being evaluated are very similar (v2.0 and v2.1). It would be better if a third dataset can be evaluated.
>
> **A3:** Although similar, MultiWOZ2.1 differs from 2.0 in that it fixes annotation errors and serves as another challenging benchmark for the DST problem. The same DST model may perform differently on these two datasets [1].
> Having said that, as responded in A1, we will add experiments on the SGD dataset when we are allowed to modify our paper.
>
> > Q4. One underlying motivation for using the last-turn annotation scheme is that such data could be easily accessed from commercial service data [1]. It can be easy to obtain the actions done by the customer service staff after a phone call. It’d be appreciated if this case can be discussed and compared to the proposed approach.
>
> **A4:** We agree that obtaining the last-turns of conversations and their associated actions might be easy in practical scenarios. However, the next action is determined by the policy learning module in the pipeline-based task-oriented dialogue system [1], whereas we focused on the DST module. Hence, although actions can be easy to obtain, it is less helpful for training a DST model.
>
>
> [1] Multi-Task Pre-Training for Plug-and-Play Task-Oriented Dialogue System, ACL’22

---

### Official Review · Reviewer_Es6J · 2023-08-05

**Soundness:** 4

**Excitement:**

4: Strong: This paper deepens the understanding of some phenomenon or lowers the barriers to an existing research direction.

**Paper Topic And Main Contributions:**

The paper proposes a novel turn-level active learning framework for Dialogue State Tracking(DST) to reduce the human annotation cost of the DST dataset. The proposed approach achieves significantly better DST performance with redcued annotation cost.

**Reasons To Accept:**

1. The experimental results of the proposed turn-level approach on the MultiWOZ dataset are significanttly better than the previous dialogue-level active learning approach.
2. The paper is well written and easy to follow.
3. The authors did comprehensive analyses with different hyperparameters and different turn selection methods.


**Reasons To Reject:**

1. The paper lacks comparisons of the training computation cost between their proposed approach and the previous dialogue-level approach. It is unclear how much computation cost is added for the improved performance (or reduced annotation cost).

**Reproducibility:**

4: Could mostly reproduce the results, but there may be some variation because of sample variance or minor variations in their interpretation of the protocol or method.

**Reviewer Confidence:**

4: Quite sure. I tried to check the important points carefully. It's unlikely, though conceivable, that I missed something that should affect my ratings.

---

> ### Author Rebuttal · Authors · 2023-08-29
>
> Dear Reviewer Es6J,
>
> We appreciate your valuable time, review and suggestions. We want to address your concerns in detail as follows.
>
> > Q1. The paper lacks comparisons of the training computation cost between their proposed approach and the previous dialogue-level approach. It is unclear how much computation cost is added for the improved performance (or reduced annotation cost).
>
> **A1:**
> Thanks for your suggestion. Using the PPTOD-base model as an example, we did a preliminary analysis of the computation cost (as shown in the table below). We will provide more analysis when we can modify our paper.
>
> Specifically, when training using the full dataset, it took about 4 hours and achieved 53.2% JGA score on MultiWOZ 2.0. When training using only the last turns, it took about 0.8 hours and achieved 43.5% JGA score. When using our proposed active learning framework and using four iterative loops, it took about 1.9 hours in total and achieved 46.8% JGA score. All experiments run on a single RTX 3090. As discussed at L585, our proposed method can achieve similar or even better DST performance with significantly reduced annotation data than the last turn baseline, with the cost of slightly increased computational cost.
>
> | **Method**                                | **% of Training data &#8595;** | **JGA &#8593;** | **RC &#8595;** | **Running time (hour) &#8595;** |
> |-------------------------------------------|------------------------|---------|--------|-------------------------|
> | PPTOD-base + Full data                     | 100%                   | 53.2    | 100    | 4                       |
> | PPTOD-base + Last Turn                     | 14.4%                  | 43.5    | 100    | 0.8                     |
> | **PPTOD-base + selected turn + ME (ours)** | 14.4%                  | 46.8    | 55.9   | 1.9                     |

---

### Official Review · Reviewer_fWKM · 2023-08-07

**Soundness:** 4

**Excitement:**

4: Strong: This paper deepens the understanding of some phenomenon or lowers the barriers to an existing research direction.

**Paper Topic And Main Contributions:**

This paper proposes a turn level active learning framework for DST that can help train a DST with minimal amount of annotation.  Novelty of the paper is that it is looking into turn level annotation rather than conversation level which is a unstudied task for DST. Their results and ablation studies shows comparable performance to traditional training with a significantly less annotated data.

**Reasons To Accept:**

•	Results are presented well with detailed ablation study, The ablation analysis and case studies provide additional insight into the effectiveness of the approach.
•	It presents a novel approach with strong performance, and clear motivation.  Turn selection based on maximum entropy showed significant improvement across both Multi-Woz dataset.
•	This is a very well written paper, coherent and easy to understand.
•	Overall, this is an effective paper with strong results.


**Reasons To Reject:**

Code can be released for other users to work on top of the current work

**Reproducibility:**

3: Could reproduce the results with some difficulty. The settings of parameters are underspecified or subjectively determined; the training/evaluation data are not widely available.

**Reviewer Confidence:**

4: Quite sure. I tried to check the important points carefully. It's unlikely, though conceivable, that I missed something that should affect my ratings.

---

> ### Author Rebuttal · Authors · 2023-08-29
>
> Dear Reviewer fWKM,
>
> We appreciate your valuable time, review and suggestions. We want to address your concerns in detail as follows.
>
> > Q1. Code can be released for other users to work on top of the current work
>
> **A1:** Thanks for your interest. We will release all code and data.

---

### Meta-Review · Area_Chair_x7E7 · 2023-09-07

**Recommendation:** 5

**Metareview:**

This paper addresses an essential yet under-explored problem in Dialogue State Tracking: minimizing the amount of turn-level annotation. The proposed active learning approach at the turn level, which is based on maximum entropy, yields performance comparable to training with a complete (and considerably larger) training corpus. Both the ablation studies and case analyses offer valuable insights. A notable limitation of this research is the additional computational cost it introduces. While the authors have present the computation study in their rebuttal, I recommend incorporating the analysis into the camera-ready version of the paper.

All reviewers agree that the paper is solid and offers value to the dialogue field.

---

### Decision · Program_Chairs · 2023-10-07

**Decision:**

Accept-Main

**Comment:**

This paper addresses an essential yet under-explored problem in Dialogue State Tracking: minimizing the amount of turn-level annotation. The proposed active learning approach at the turn level, which is based on maximum entropy, yields performance comparable to training with a complete (and considerably larger) training corpus. Both the ablation studies and case analyses offer valuable insights. A notable limitation of this research is the additional computational cost it introduces. While the authors have present the computation study in their rebuttal, I recommend incorporating the analysis into the camera-ready version of the paper.

All reviewers agree that the paper is solid and offers value to the dialogue field.